# Preparation, Characterization, and Pharmacokinetic Evaluation of Imperatorin Lipid Microspheres and Their Effect on the Proliferation of MDA-MB-231 Cells

**DOI:** 10.3390/pharmaceutics10040236

**Published:** 2018-11-16

**Authors:** Xinli Liang, Xulong Chen, Guowei Zhao, Tao Tang, Wei Dong, Chunyan Wang, Jing Zhang, Zhenggen Liao

**Affiliations:** Key Laboratory of Modern Preparation of Traditional Chinese Medicine, Jiangxi University of Traditional Chinese Medicine, Nanchang 330004, China; Paln7@163.com (X.L.); cxl0517@163.com (X.C.); weiweihaoyunqi@163.com (G.Z.); marryrose1013@163.com (T.T.); sober96@foxmail.com (W.D.); doscat@163.com (C.W.); evens_zhang@163.com (J.Z.)

**Keywords:** imperatorin, lipid microsphere, response surface methodology, pharmacokinetic

## Abstract

Imperatorin is a chemical compound belonging to the linear furanocoumarins. Imperatorin is attracting considerable attention because of its antitumor, antibacterial, anti-inflammatory, and anticoagulant activities, inhibition of myocardial hypertrophy, and other pharmacological efficacies. However, imperatorin has limited water solubility and has better lipid solubility; thus, we decided to design and synthesize imperatorin lipid microspheres to optimize the preparation conditions. The aim was to develop and formulate imperatorin lipid microspheres through nanoemulsion technology and apply the response surface–central composite design to optimize the imperatorin lipid microsphere formulation. The influence of the amounts of egg lecithin, poloxamer 188, and soybean oil for injection on the total percentage of the oil phase was investigated. The integrated effect of dependent variables, including particle size, polydispersity index, zeta potentials, drug loading, and encapsulation efficiency, was investigated. Data of overall desirabilities were fitted to a second-order polynomial equation, through which three-dimensional response surface graphs were described. Optimum experimental conditions were calculated by Design-Expert 8.06. Results indicated that the optimum preparation conditions were as follows: 1.39 g of egg lecithin, 0.21 g of poloxamer 188, and 10.57% soybean oil for injection. Preparation of imperatorin lipid microspheres according to the optimum experimental conditions resulted in an overall desirability of 0.7286, with the particle size of 168 ± 0.54 nm, polydispersity index (PDI) of 0.138 ± 0.02, zeta potentials of −43.5 ± 0.5 mV, drug loading of 0.833 ± 0.27 mg·mL^−1^, and encapsulation efficiency of 90 ± 1.27%. The difference between the observed and predicted values of the overall desirability of the optimum formulation was in the range from 2.4% to 4.3%. Subsequently, scanning electron microscopy was used to observe the micromorphology of the imperatorin lipid microspheres, showing round globules of relatively uniform shape and sizes within 200 nm. The effect of imperatorin lipid microspheres on MDA-MB-231 proliferation was investigated by the MTT method. Furthermore, pharmacokinetics in Sprague-Dawley rats was evaluated using orbital bleeding. A sensitive and reliable liquid chromatography with the high-performance liquid chromatography (HPLC) method was established and validated for the quantification of imperatorin in rat plasma samples. The data were calculated by DAS (drug and statistics) Pharmacokinetic Software version 3.3.0 (Version 3.3.0, Shanghai, China). Results demonstrated that imperatorin lipid microspheres can significantly enhance the bioavailability of imperatorin and can significantly inhibit MDA-MB-231 cell proliferation. In conclusion, our results suggested that the response surface–central composite design is suitable for achieving an optimized lipid microsphere formulation. Imperatorin lipid microspheres can improve the bioavailability of imperatorin and better inhibit the proliferation of MDA-MB-231 cells as compared to imperatorin alone.

## 1. Introduction

Imperatorin is a chemical compound belonging to the linear furanocoumarins and it is mainly extracted and isolated from the traditional herbal medicine of *Angelica dahurica*. Modern pharmaceutical studies have identified that imperatorin has good biological activity, including analgesic, antibacterial, anti-inflammatory, vessel dilating, and CYP450-inhibiting effects [1]. Many studies have suggested that imperatorin had certain antitumor efficacies, such as an inhibiting effect on a human hepatocellular carcinoma cell line, a human breast cancer cell line, a human cervical carcinoma cell line, a human osteosarcoma cell line, and other metastatic tumor cells. The mechanism mainly included a downregulating effect on mitochondrial MCl-1 protein expression [2,3,4,5]. Jakubowicz-Gil showed that imperatorin combined with quercetin could effectively inhibit the proliferation of tumor cells and induce the cells’ apoptosis [6,7,8]. These findings suggested that imperatorin is a potential anticancer drug with a good application prospect. So, it is obvious that the compound has great potential to be developed as a pharmaceutical formulation for subsequent clinical assessment. But, Imperatorin belongs the BCS II classfication system. It shows a relatively low bioavailability because of its poor water solubility [9]. So, it is difficult to prepare an ideal oral pharmaceutical preparation [10]. It is more difficult to be developed as an injection. Research reports ultradeformable liposomes as a novel vesicular carrier could increase skin permeation efficiency of imperatorin [11], and Jingjing Pan prepared imperatorin sustained-release tablets to lower blood pressure [12]. There is no injection of imperatorin reported. In this study, we took advantage of imperatorin’s characteristics of poor solubility in water and good fat solubility and dissolved it in an oil-based injection, selected lecithin as an emulsifier, and used nanoemulsification technology to prepare imperatorin lipid microspheres.

Lipid microsphere (LM) has been recently used as intravenous (i.v.) carriers for drugs, especially for those drugs that have enough solubility in oil while have poor solubility in water. It is difficult for this kind of drugs to prepare injection preparations. The LM, with a diameter of 0.2 microns, is composed of soybean oil and lecithin, can carry lipophilic or hydrophilic drugs, drugs are usually incorporated into the oil phase or into the interface of oil phase in LM, so they are presumed to avoid or reduce local or blood vessel irritation. As a lipophilic drug carrier, it has the following advantages of targeting positioning, improving the solubility and stability of drugs, reducing the adverse reactions, etc. [13,14]. 

Uniform design and orthogonal design are two kinds of experimental design methods that are widely used in the research of pharmaceutical preparations of Chinese herbal medicines [15]. However, the uniform design and orthogonal design optimization method is constrained by the linear model, as it can only point out the direction of factor values and is unable to find extremes, and the deviation between the measured and predicted values is larger under the optimum preparation conditions [16]. Response surface methodology (RSM) is a combination of mathematical and statistical techniques, which has the characteristics of requiring fewer tests and having higher test accuracy. It is also more simplified and comprehensive than orthogonal design and uniform design. In the process of optimization, practical research mainly focuses on central composite design (CCD) under RSM [17]. Because CCD is very practically suitable for comparing experimental methodologies with theoretical models [18], and it includes not only the effects of interaction of the variables but also the overall effects of the parameters in the process [19], it is often used in the optimization method for the preparation of new technologies.

Breast cancer is one of the most common female cancers in the world. It is still associated with high morbidity and mortality. At present, chemotherapy and surgery are the most important methods of treating breast cancer. Imperatorin is a single component of some traditional Chinese medicines, which has the characteristics of high efficacy and low toxicity. Previous studies have showed that imperatorin has an antitumor effect [20,21], and it has a strong inhibitory effect on MDA-MB-231 cells [22]. However, because of its physical and chemical properties, its druggability is very low. In order to increase its druggability and to exert its antitumor effect, the optimization of the preparation and formulation of imperatorin lipid microspheres was accomplished; furthermore, the pharmacokinetics of imperatorin in rats was investigated, and the respective effects of imperatorin and imperatorin lipid microspheres on MDA-MB-231 cell proliferation were also compared in the study.

## 2. Materials and Methods

### 2.1. Materials

#### 2.1.1. Chemicals and Drugs

Imperatorin was purchased from the National Institutes for Food and Drug Control (batch: 110826–200511, Beijing, China). Soybean oil for injection (long-chain triglyceride, LCT) and medium-chain fatty acid glyceride for injection (MCT) were purchased from Tieling North Asia Medicinal Oil Co. Ltd. (Tieling, China). Egg lecithin was purchased from Dongshang biotechnology Co. Ltd. (Shanghai, China). Glycerol for injection was purchased from the Jiangxi Benefit Spectrum Health Pharmaceutical Division (Nanchang, China). Poloxamer 188 was purchased from Shanghai Changsheng Technology Co. Ltd. (Shanghai, China). The reagents were chromatographically and analytically pure.

Fetal calf serum (FCS) and RPMI1640 were purchased from Hyclone (Thermo Fisher Scientific). Penicillin and streptomycin solutions (10,000 U/mL penicillin and 10,000 mg/mL streptomycin) were purchased from Solarbio (Beijing Solarbio Science & Technology Co., Ltd., Beijing, China). Nonessential amino acids were obtained from Sigma Chemical Co. (St. Louis, MA, USA). Trypsin–EDTA solution (0.25% (*w*/*w*) trypsin/1 mM EDTA) was supplied by Gibco Laboratories (Life Technologies Inc., Carlsbad, CA, USA). MTT cell proliferation and Cytotoxicity Detection Kit (batch: 20170613) were purchased from Jiangsu KeyGEN BioTECH Corp., Ltd. (Nanjing, China). MDA-MB-231 cells were purchased from the cell bank of the Chinese Academy of Sciences.

#### 2.1.2. Animals

Male Sprague-Dawley (SD) rats were purchased from Slack King Experimental Animal Center in Hunna (Hunan, Changsha, China). Before the experiment, all rats were housed in an environmentally controlled room (25 ± 2 °C and relative air humidity 52 ± 20%) with free access to food and water. All animal experiments were approved by the Animal Center Committee of Jiangxi University of Traditional Chinese Medicine, all of which were conducted in full compliance with the local, national, ethical, and regulatory principles.

### 2.2. Methods

#### 2.2.1. Imperatorin Lipid Microsphere Preparation

Imperatorin lipid microspheres were prepared with a high-speed shearing and high-pressure homogenization method, as described previously [23,24]. Imperatorin was dissolved in the oil phase, which was composed of egg yolk lecithin, LCT, and MCT. The water phase was composed of glycerol, sodium oleate, and poloxamer 188. The oil phase and water phase were both heated to 70 °C, and then the hot oil phase was added to the water phase and stirred in a high-speed shearing homogenizer for 10 min at a revolution speed of 19,000 rpm to obtain the colostrum. Thereafter, the colostrum was circulated six times at 600 bar in the homogenizer, after which imperatorin lipid microspheres were obtained.

#### 2.2.2. Measurement of Size, PDI, and Zeta Potential of Imperatorin Lipid Microsphere

The average particle size, PDI, and zeta potential of the lipid microspheres were measured while using a Malvern laser particle size analyzer (Mal-vern, UK). Samples were diluted appropriately with double-steamed water for the measurements, and zeta potential measurements were detected at 25 °C.

#### 2.2.3. Scanning Electron Microscopy (SEM)

The morphologies of the imperatorin lipid microspheres and blank lipid microspheres were observed using a FEIQuanta 250 SEM (FEI Corporation, Hillsboro, OR, USA). After dilution with double-steamed water, a drop of solution was placed on the sample stand, was drained naturally, and was sprayed for observation.

#### 2.2.4. Determination of Drug Loading and Encapsulation Efficiency

Encapsulation efficiency was determined by ultra-high-speed centrifugation. In addition, the drug loading and encapsulation efficiency of imperatorin was determined following the solubilization of carriers in methanol and analysis by the high-performance liquid chromatography (HPLC) method. The mobile phase consisted of methanol and double-distilled water (80:20, *v*/*v*). A volume of 20 μL of sample was injected and the flow rate was 1 mL/min. The column temperature was maintained at 25 °C and the detection wavelength was set at 330 nm [25].

The drug loading was calculated according to the standard curve [26]
Encapsulation efficiency (%) = (C_o_V_o_ − C_w_V_w_)/C_o_V_o_ × 100%
Drug loading = (C_a_W_b_)/W_a_

#### 2.2.5. RSM Design and Optimization of Imperatorin Lipid Microsphere Preparation Conditions

RSM was developed to acquire the optimal preparation conditions by establishing the relationships between the variables and the response.

Based on the single factor test results of preliminary experiments and our previous studies, three formulation parameters, namely the amounts of egg lecithin (A), poloxamer 188 (B), and soybean oil for injection, accounting for the total percentage of the oil phase (C), were identified as the key factors that are responsible for the particle size (Y_1_), polydispersity index (Y_2_), zeta potential (Y_3_), drug loading (Y_4_), and encapsulation efficiency (Y_5_). The range and levels of the three independent variables used in this study is summarized in Table 1. The central composite design experiments were carried out in a randomized order, which included six repeated experiments to eliminate the system error. Dependent variables or responses were transformed into desirabilities mathematically by Hassan’s method. Overall desirability was calculated from the geometric mean of five desirabilities of each formulation. In this method, we set the best value as 1 and the worst value as 0, and all desirabilities are normalized from 0 to 1.

The formula to calculate the overall desirability was expressed as follows [27]:OD = (d_1_d_2_d_3_d_4_d_5_)^1/5^
d_min_ = (Y_max_ − Y_i_)/(Y_max_ − Y_min_)
d_max_ = (Y_i_ − Y_min_)/(Y_max_ − Y_min_)
where d is the overall desirability of each independent variable; d_1_, d_2_, d_3_, d_4_, and d_5_ are the overall desirabilities of particle size, particle size distribution, zeta potential, drug loading, and encapsulation efficiency, respectively; Y is the determination value of each independent variable (i = 1, 2, 3, 4, 5); and, Y_max_ and Y_min_ are the maximum and minimum, respectively, of each independent variable in all the tests.

Design-Expert 8.0 software was used to analyze the experimental data of overall desirabilities, perform multiple regressions to obtain the coefficients of the cubic polynomial model, and to obtain the three-dimensional response surface graphs. The quality of the fitted model was expressed by the coefficient of determination R^2^, and its statistical significance was determined by the *F*-test.

#### 2.2.6. Pharmacokinetics and Statistical Analysis

Six male Sprague-Dawley rats were given imperatorin lipid microspheres (1 mg·mL^−1^) by the means of IV at a dose of 5 mg/kg. Orbital blood samples (200 μL) were collected at 2, 5, 10, 15, 20, 30, 45, 60, 90, 120, 180, 240, and 360 min after administration. Six male Sprague-Dawley rats were given imperatorin suspensions (dissolved in appropriate amount DMSO) by the means of intragastric administration at a dose of 50 mg/kg. Orbital blood samples (200 μL) were collected at 2, 10, 20, 30, 40, 50, 60, 90, 120, 180, 240, 360, 480, and 600 min after oral administration. Blood samples were placed in heparinized tubes and immediately centrifuged in a centrifuge tube coated with sodium heparin at 4000 rmp·min^−1^ at 4 °C for 10 min. The supernatant was taken and stored at −80 °C until further analysis.

Plasma samples were treated by a liquid-liquid extraction method. 100 μL of plasma samples (containing internal standard oxypeucedanin) and 900 μL of methanol were placed in a 1.5 mL of Eppendorf tube. Samples were then vortex-mixing for 3 min and centrifuged for 30 min at 16,000 rmp. Approximately, 1 mL of supernatant was placed into another clean tube and filtered with 0.22 μm filter.

Samples were analysed by using a HPLC (Aglient 1260, Agilent Technology, Santa Clara, CA, USA). The chromatographic conditions are as follows: Phenomenex-C_18_ (4.6 mm × 250 mm, 5 μm); the mobile phase consisted of methanol and double distilled water (80:20, *v*/*v*), A volume of 20 μL of sample was injected, and the flow rate was 1 mL/min. The column temperature was maintained at 25 °C and the detection wavelength was set at 330 nm [21].

Pharmacokinetic parameters of imperatorin after intravenous injection of imperatorin lipid microsphere were calculated by Software version 3.3.0 (Shanghai, China).

#### 2.2.7. Effect of Imperatorin and Imperatorin Lipid Microspheres on MDA-MB-231 Cell Proliferation

MDA-MB-231 cells were cultured in medium containing RPMI1640 (10% fetal bovine serum, 1% nonessential amino acids, 1% l-glutamine, 100 U/mL penicillin–streptomycin). The cells were maintained at 37 °C in an atmosphere containing 5% CO_2_ at 95% relative humidity. The medium was changed every other day during cell growth and differentiation. The cells could be used in the experiments when they had grown to 80–90%. The cells were seeded onto 96-well plates at a density of 5 × 10^4^ and supernatant was discarded after 24 h of growth. Different concentrations of imperatorin and imperatorin lipid microspheres were added to the cells and cells were subsequently cultured for different amount of time (24, 48, and 72 h), according to the instructions of the MTT cell proliferation and cytotoxicity test kit, to study the effect of imperatorin and imperatorin lipid microspheres on MDA-MB-231 cell proliferation.
$$Inhibition\;of\;cell\;proliferation(\%)=\frac{Control\;group\;OD-Drug\;group\;OD}{Control\;group\;OD}\times 100\%\;$$

### 2.3. Data Analysis

All experimental data in this experiment were expressed as the mean ± standard error. Data analyses were performed by using the DAS 3.3.0 pharmacokinetic program (Chinese Pharmacology Society, Shanghai, China). All statistical analyses were analyzed using Student’s *t*-test.

## 3. Results and Discussion

### 3.1. Central Composite Design of Response Surface Methodology

The experimental data are summarized in Table 2. The statistical significance of the regression model was analyzed by *p*-value and F-test, and the analysis of variance (ANOVA) for the response surface quadratic model is shown in Table 3, in which the *p*-values < 0.01 implied the model was very significant, and *p*-values < 0.05 suggested the model term was significant. The *p*-value for the “Lack of Fit” test was 1.78, indicating that the quadratic model was adequate.

After statistical processing and fitting, multiple regression equations were obtained, as follows:

Final equation in terms of coded factors:OD = 0.51 + 0.082A − 0.081B − 0.011C − 0.27AB − 9.205E − 003AC + 9.205E − 003BC − 0.096A^2^ − 0.097B^2^ − 0.035C^2^

Final equation in terms of actual factors:OD = 0.50837 + 0.081881A − 0.081132B − 0.010991C − 0.27387AB − 9.20457E − 003AC + 9.20457E − 003BC − 0.095557A^2^ − 0.096566B^2^ − 0.035043C^2^

The analysis of fitting is shown in Table 4.

The above regression equations quantitatively described the relationship between the three independent variables (A, B, and C) on index and the overall desirability. The adjusted R^2^ for the predictive model is 0.8918 and the statistical test results of equation parameters are summarized in Table 4. It is revealed that the experimental results adequately fitted the selected regression equations. The “Adj R-Squared” value of 0.7944 is not as close to the “Pred R-Squared” value of 0.4031, as one might normally expect. This may indicate a possible problem or a large block effect with the model and/or data. Things to consider are response transformation, model reduction, outliers, and so forth. “Adeq Precision” measures the signal-to-noise ratio and a ratio greater than 4 is desirable. The ratio of 9.418 indicates an adequate signal. This model can be used to navigate the design space. It can be used predictively the obtain response value of a random formula within the range and level of independent factors by regression equations.

To better comprehend the predictive three-dimensional graphs of the models in the results, the response surface diagrams of imperatorin lipid microspheres were created, as shown in Figure 1. The optimum formulation conditions were as follows: the amount of egg lecithin is 1.39 g, the amount of poloxamer 188 is 0.21 g, and the amount of soybean oil for injection is 10.57 g.

The recommended optimum conditions were used to test the suitability of the model equation for predicting the optimum response values. According to the model equation, the RSM optimization approach was used to determine the optimum conditions. Three batches of imperatorin lipid microspheres were prepared according to the optimized formulation. Table 5 listed the optimum ranges and the experimental and predicted values for the response variables under the test conditions, and the calculated percentage prediction error. As seen from Table 6, the prediction error of the response variables was found to vary between 2.4% and 4.3%. The results of the verifying experiments were very close to the predicted values that were obtained from the optimization analysis using the desirability function with low prediction error, suggesting that the optimization was reasonable and reliable.

### 3.2. Drug Loading and Encapsulation Efficiency

The drug loading and encapsulation efficiency of three batches is reported in Table 7.

### 3.3. Particle Size and Zeta Potential Measurements

The results of particle size and zeta potential are shown in Table 8. From the results, we can see that the imperatorin lipid microspheres have the traits of small size and narrow size distribution.

### 3.4. Scanning Electron Microscopy (SEM)

SEM images of the imperatorin lipid microspheres are shown in Figure 2. The imperatorin lipid microspheres were small homogenous vesicles with a bilayer lipid membrane.

### 3.5. Pharmacokinetic Study

A dose of 5 mg·kg^−1^ of imperatorin lipid microspheres was injected intravenously into rat subjects. A dose of 50 mg·kg^−1^ imperatorin was given by the means of intragastric administration. The pharmacokinetic parameters were calculated by DAS software. The mean plasma concentration–time curves are shown in Figure 3. The major pharmacokinetic parameters are listed in Table 9.

When compared with the oral administration of imperatorin, the AUC_(0–t)_ of imperatorn lipid microspheres significantly increased and the peak (maximum) plasma concentration (C_max_) of imperatorin lipid microspheres (77.46 ± 23.82 mg·L^−1^) is much higer than that of imperatiorin (5.75 ± 1.59 mg·L^−1^). Upon IV administration at a dose of 5 mg·kg^−1^, the time to peak (maximum) concentration (T_max_) was at 2 min after intravenous administration of imperatorn lipid microspheres in rats, and the time to peak (maximum) concentration (T_max_) was at 45 min after oral administration of imperatorin in rats, indicating that imperatorin lipid microspheres could be quickly detected in plasma. While imperatorin lipid microspheres was shown to have a short half-life (t1/2 = 1.00 ± 0.40 h) and a clearance of 0.04 ± 0.01 L·h^−1^·kg^−1^ than that of an oral imperatiorin (t1/2 = 4.02 ± 1.09 h, CLz/F = 2.63 ± 0.98). The short half-life suggests that imperatorin lipid microspheres should be quickly metabolized in vivo and they should have a short duration of efficacy. The result suggested that we should investigate prolonging the half-life of imperatorin lipid microspheres.

### 3.6. Effect of Imperatorin and Imperatorin Lipid Microspheres on MDA-MB-231 Cell Proliferation

The results showed that the inhibition mediated by imperatorin and imperatorin lipid microspheres on MDA-MB-231 cell proliferation all had a positive correlation with the culture time (Figure 4). With increasing concentration of imperatorin or imperatorin lipid microspheres, the degree of inhibition of MDA-MB-231 cell proliferation improved correspondingly. When compared with the effect of imperatorin, the imperatorin lipid microspheres group had a stronger inhibitive effect on MDA-MB-231 cell proliferation than that of imperatorin.

## 4. Conclusions

The lipid microsphere is a good candidate for drug loading because of its safety, stability, and good biocompatibility, especially for those drugs with low solubility.

The central composite design–response surface method is an optimal design method that is used in the optimization of formulations due to its relatively small number of experiments required and high precision. According to the surface change, the three-dimensional effect of the surface chart could directly respond to the influence of factors on the survey index. Based on the overlying of better conditions chosen by multiple effects, the range of better conditions can be further reduced.

The optimal formulation was: egg lecithin—13.9 g, poloxamer 188—2.1 g, and soybean oil—105.7 g. The particle size was 168 ± 1.73 nm, polydispersity index (PDI) was 0.138 ± 0.02, zeta potential was −43.5 ± 0.5 mV, drug loading was 0.833 ± 0.027 mg/mL, and the encapsulation efficiency was 90 ± 1.27%. The results showed that imperatorin lipid microspheres could significantly inhibit MDA-MB-231 cell proliferation. However, pharmacokinetic study of the imperatorin lipid microspheres showed that the half-time of imperatorin was very short. Thus, further studies should be conducted on how to increase its half-life duration and improve the residence time in blood.

## Figures and Tables

**Figure 1 pharmaceutics-10-00236-f001:**
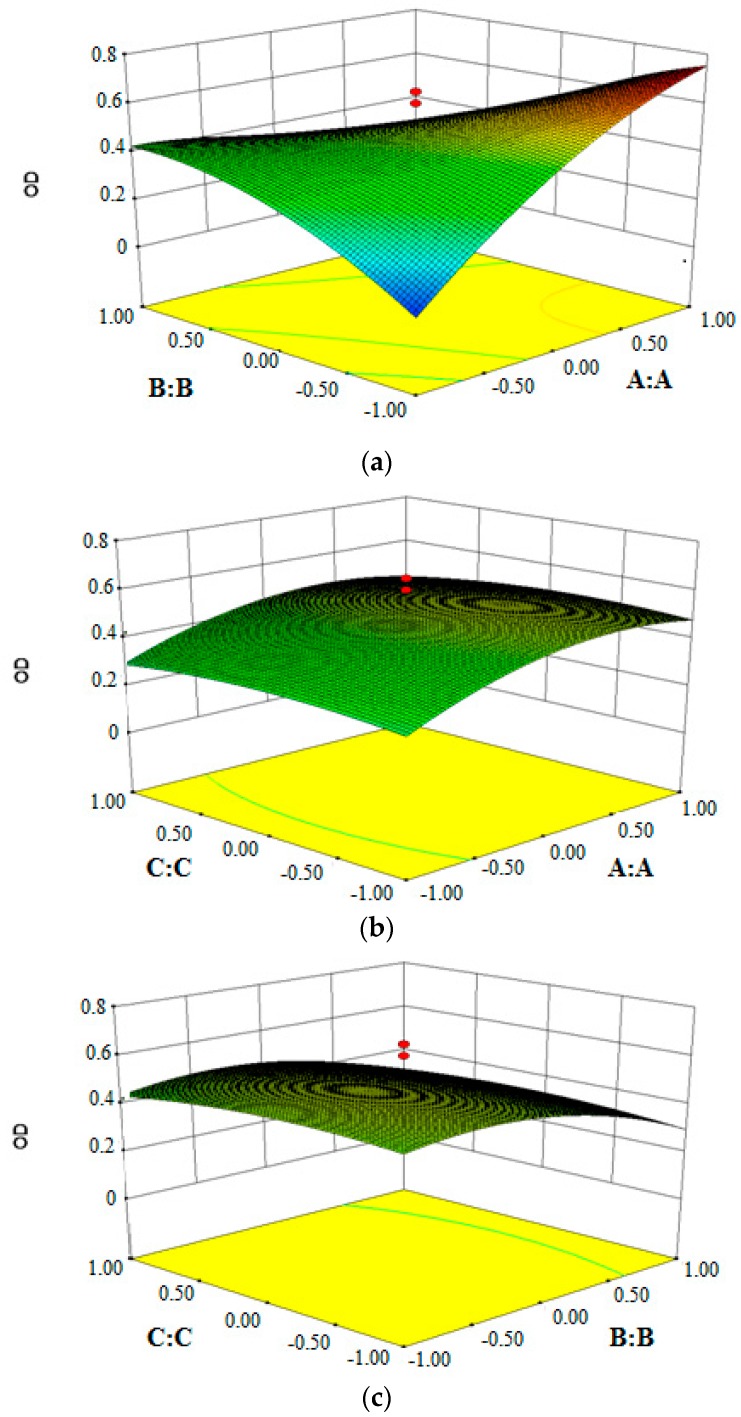
Response surface plot. (**a**) The effect of interaction of the egg lecithin and poloxamer 188 on OD value; (**b**) The effect of interaction of the egg lecithin and LCT/oil phase ratio on OD value; (**c**) The effect of interaction of poloxamer 188 and LCT/oil phase ratio on OD value.

**Figure 2 pharmaceutics-10-00236-f002:**
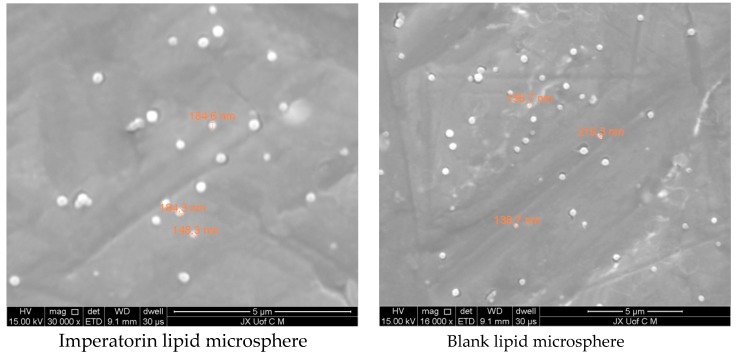
Scanning electron microscopy of imperatorin lipid microspheres.

**Figure 3 pharmaceutics-10-00236-f003:**
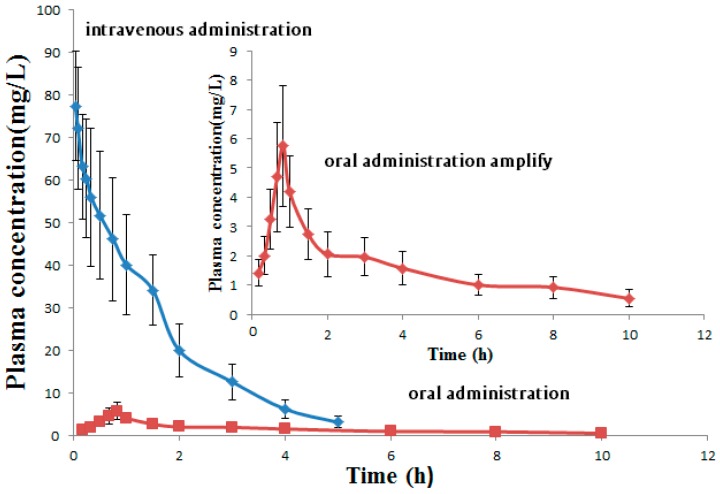
Mean plasma concentration–time curves after intravenous administration of 5 mg/kg imperatorin lipid microspheres and oral administration of 50 mg·kg^−1^ imperatorin in rats.

**Figure 4 pharmaceutics-10-00236-f004:**
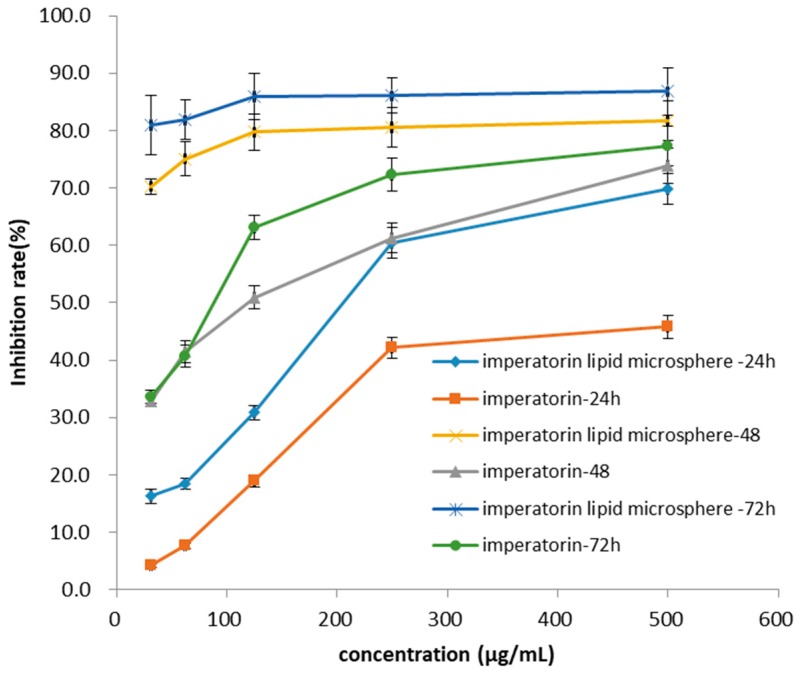
Inhibition cures of imperatorin and imperatorin lipid microspheres against MDA-MB-231.

**Table 1 pharmaceutics-10-00236-t001:** Levels and code of variables chosen for the central composite design.

Factors	Code	Range and Levels				
		−1.732	−1	0	1	1.732
egg lecithin	A	1	1.11	1.25	1.39	1.5
Poloxamer 188	B	0.1	0.21	0.35	0.49	0.6
Soybean oil/oil phase	C	0	10.57	25.00	39.43	50

**Table 2 pharmaceutics-10-00236-t002:** Variables and observed responses in the central composite design for imperatorin lipid microspheres.

No. Levels of Independent Factors’ Responses									
	A	B	C	Y_1_	Y_2_	Y_3_	Y_4_	Y_5_	OD
1	1.11	0.21	39.43	177	0.148	−43.4	6.58	89%	0
2	1.25	0.35	25.00	172	0.131	−44.1	7.59	90%	0.4835
3	1.11	0.49	10.57	169	0.168	−47.0	8.27	89%	0.5195
4	1.25	0.35	25.00	172	0.128	−43.7	7.72	88%	0.4582
5	1.25	0.35	25.00	161	0.138	−45.0	7.29	89%	0.4562
6	1.39	0.49	39.43	164	0.097	−38.7	6.93	81%	0
7	1.25	0.35	25.00	165	0.148	−43.5	7.23	90%	0.4013
8	1.25	0.35	0	167	0.129	−43.4	7.16	84%	0.3693
9	1.11	0.21	10.57	193	0.122	−45.2	9.43	91%	0
10	1.0	0.35	25.00	201	0.132	−43.9	9.02	89%	0
11	1.5	0.35	25.00	168	0.134	−43.8	7.33	88%	0.4629
12	1.39	0.49	39.43	177	0.183	−41.9	9.14	89%	0.5391
13	1.39	0.49	10.57	154	0.116	−44.5	8.28	81%	0
14	1.11	0.49	39.43	165	0.094	−42.4	7.29	88%	0.4037
15	1.39	0.21	10.57	168	0.138	−43.5	8.33	90%	0.7286
16	1.25	0.35	25.00	164	0.120	−43.5	7.76	89%	0.6020
17	1.25	0.35	50.00	176	0.136	−44.7	7.88	90%	0.4567
18	1.25	0.35	25.00	170	0.129	−43.1	10.42	90%	0.6491
19	1.25	0.1	25.00	196	0.097	−41.5	10.92	92%	0.4569
20	1.25	0.6	25.00	170	0.096	−40.5	9.27	93%	0

**Table 3 pharmaceutics-10-00236-t003:** Statistical analysis of variance for the experimental results.

Source	Sum of Squares	df	Mean Square	*F* Value	*p*-Value Prob > 7
Model	1.05	9	0.12	9.16	0.0009 *
A-A	0.094	1	0.094	7.36	0.0218
B-B	0.092	1	0.092	7.23	0.0228
C-C	1.691 × 10^−3^	1	1.691 × 10^−3^	0.13	0.7233
AB	0.60	1	0.60	108.14	<0.0001
AC	6.778 × 10^−4^	1	6.778 × 10^−4^	0.053	0.8223
BC	6.778 × 10^−4^	1	6.778 × 10^−4^	0.053	0.8223
A^2^	0.14	1	0.14	11.28	0.0073
B^2^	0.15	1	0.15	11.52	0.0068
C^2^	0.019	1	0.019	1.52	0.2462
Residual	0.13	10	0.013		
Lack of Fit	0.082	5	0.016	1.78	0.2713
Pure Error	0.046	5	9.174 × 10^−3^		
Cor Total	1.18	19			

Note: * *p* < 0.01 the model is very significant

**Table 4 pharmaceutics-10-00236-t004:** The results of fitting second-order equations.

Item	Data	Item	Data
Std. Dev.	0.11	R-Squared	0.8918
Mean	0.35	Adj R-Squared	0.7944
C.V.%	32.32	Pred R-Square	0.4031
PRESS	0.70	Adeq Precision	9.418

**Table 5 pharmaceutics-10-00236-t005:** Constraints of factors and responses for optimization.

Name	Goal	Lower Limit	Upper Limit	Lower Weight	Upper Weight	Important
A: egg lecithin	is in range	1.0	1.5	1	1	3
B: poloxamer 188	is in range	0.1	0.6	1	1	3
C: LCT/oil phase ratio	is in range	0	50	1	1	3
Responses: OD	maximize	0	0.7286	1	1	3

**Table 6 pharmaceutics-10-00236-t006:** The experimental and values for response (OD) along with percentage prediction error observed for the optimum test conditions.

Batch	A	B	C	OD		
				Predicted Value	Experimental Value	Percent Prediction Error
20171101	1.39	0.21	10.57	0.7580	0.7286	3.8%
20171102	1.39	0.21	10.57	0.7580	0.7395	2.4%
20171103	1.39	0.21	10.57	0.7580	0.7251	4.3%

**Table 7 pharmaceutics-10-00236-t007:** Drug loading and encapsulation efficiency of imperatorin lipid microspheres ($\bar{x}\pm s$, *n* = 3).

Batch	Drug Loading (mg/mL)	Encapsulation Efficiency (%)
20171101	0.815	90.3
20171102	0.836	91.2
20171103	0.859	88.7
Mean	0.833 ± 0.027	90.0 ± 1.27

**Table 8 pharmaceutics-10-00236-t008:** Zeta potential and particle size of imperatorin lipid microspheres ($\bar{x}\pm s$, *n* = 3).

Batch	Zeta Potential (mv)	Particle Size (nm)	PDI
Carrier	−44.9 ± 1.20	154 ± 4.92	0.157 ± 0.04
20171101	−43.1	169	0.114
20171102	−44.1	165	0.159
20171103	−43.5	169	0.142
Mean	−43.5 ± 0.50	168 ± 1.73	0.138 ± 0.02

**Table 9 pharmaceutics-10-00236-t009:** Main pharmacokinetic parameters of imperatorin after intravenous administration of 5 mg·kg^−1^ imperatorin lipid microspheres and oral administration of 50 mg·kg^−1^ imperatorin in rats (mean ± SD, *n* = 6).

Parameter	Unit	Route of Administration	
		Intravenous Injection	Oral Administration
AUC(0–t)	mg/L·h	116.71 ± 38.72 **	15.92 ± 5.10
AUC(0–∞)	mg/L·h	121.24 ± 40.01 **	19.04 ± 6.57
AUMC(0–t)	h·h·mg/L	160.74 ± 60.78 **	56.13 ± 18.01
AUMC(0–∞)	h·h·mg/L	189.92 ± 70.59 **	105.49 ± 31.13
MRT(0–t)	h	1.38 ± 0.41 **	3.53 ± 1.28
MRT(0–∞)	h	1.57 ± 0.51 **	5.54 ± 1.95
t1/2z	h	1.00 ± 0.40 **	4.02 ± 1.09
Tmax	h	0.03 ± 0.01 **	0.83 ± 0.24
CLz/F	L/h/kg	0.04 ± 0.01 **	2.63 ± 0.98
Cmax	mg/L	77.46 ± 23.82 **	5.75 ± 1.59

** *p* < 0.01, compared with oral imperatorin.

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
