# Peer review of "Preparation, Characterization, and Pharmacokinetic Evaluation of Imperatorin Lipid Microspheres and Their Effect on the Proliferation of MDA-MB-231 Cells"

_pharmaceutics, 2018, doi:10.3390/pharmaceutics10040236_

Round 1

Reviewer 1 Report

This manuscript reports results the preparation, characterization, and pharmacokinetic evaluation of imperatorin lipid microsphere. Although several related studies were performed to support authors’ conclusion, overall descriptions of this manuscript seem to be poor. In addition, some comparison studies remain missing. There are some questions and/or suggestions to improve the quality of the manuscript.

1. In the introduction, the authors should indicate the previous formulation approaches of imperatorin in the literatures.

2. The authors investigated the pharmacokinetics of imperatorin after intravenous bolus injection of imperatorin lipid microsphere formulation. However, authors should demonstrate to show the improved pharmacokinetics of imperatorin with imperatorin lipid microsphere formulation, compared to imperatorin solution with proper vehicle. There are several previous reports for imperatorin pharmacokinetics such as Ngo et al., (2017, J Chromatogra. B) entitled “Simultaneous determination of imperatorin and its metabolite xanthotoxol in rat plasma and urine by LC–MS/MS and its application to pharmacokinetic studies”

Authors list several pharmacokinetic parameters in Table 9. However, a detail description is needed. Normally, VRT (i.e., the second moment parameter) is not really important.

3. In the 3.5. section, authors described the pharmacokinetic of imperatorin using term of “ imperatorin lipid 269 microsphere”  However, authors determined the level of imperatorin, not the imperatorin lipid 269 microsphere. That definition may get confused to the reader.

4. Figure 4 and Figure 5 need to indicate the standard deviation (SD).

5. Figure 3 needs to have control group.

6. Table 8 need to indicate the zeta potential and size distribution information of the carrier (lipid microsphere without imperatorin.

7. Methodology sections should be rewriten extensively. For examples, the HPLC method and LLE extraction method section were poorly described.

8. Table 9 need to have significant digits in a consistent manner.

9. Figure 2 seems to be unnecessary because Table 8 has same information.

Reviewer 2 Report

Dear the Editor

Imperatorin is a synthetic coumarin conjugated to furan, that is originally found in the traditional herbal medicine of Angelica dahurica with anti-proliferative property. Due to the hydrophobicity of imperatorin, Liang X et al examined the preparation of imperatorin lipid microsphere based on a response surface-central composite model. As variables, these authors examined the content of egg lecithin amount (A), poloxamer188 (B), soybean oil (C), respectively. In this study, they found that particle size (Y1), polydispersity index (Y2), zeta potential (Y3), drug loading (Y4), encapsulation efficiency (Y5) have significant effect on the formation of imperatorin lipid microsphere. As expected, these authors reported that the estimated and observed values showed in good agreement. In order to examine the biological characteristics, the effect of imperatorin lipid microsphere on the proliferation of MDA-MB-231 cells and the pharmacokinetics of this compound in SD rat plasma were tested. It is concluded that this estimation model is suitable for the optimization of imperatorin lipid microsphere preparation that increases its biological activity.

Major points:

Although these authors assumed to describe the potential usefulness of imperatorin when it is formulated as lipid microsphere, the manuscript seemed rather ill-organized. For example, Table 1 is not properly cited in the text. It is also recommended that manuscript should be English-edited before submission.

Reviewer 3 Report

Overall Comments:

Overall the manuscript is well drafted. It would be highly interesting for the readers if the authors can comment on the concentrations of imperatorin that was used for the PK study in rats and the concentrations of imperatorin which was used to the %inhibition.

It would also be of value to add clinical perspective in the conclusion to better understand the application.

Specific Comments:

Line 51: "Many research showed" can be changed to "Many studies suggested that "  

Line 57: "These suggested that" can be changed to "Taken together imperatorin"

Line 57: "and have" change to "and has"

Line 59-60: What is the BCS classification of this compound? If there is any.

Figure 1: Can the authors increase the size of these ?

Line 250: Upper case Table and Figure just to be consistent through out the manuscript.

Line 261:Poor solubility in what?  "Aqueous media?" If there are other studies which did this, the authors compare bioavaiablity with those studies.  

Line 266: Were there other studies done to characterize the pharmacokinetics of this compound ? If so, how do these parameters compare with those studies ? Is there increased benefit of this formulation compared to previously done studies or other formulations ?

line 271: Can the authors be consistent with the decimal places ? Sometimes its 2 decimal point and other places its 3 decimal points.

Line 271: Check parentheses for clearance

Figure 4: Can the authors change the pharmacokinetic plot to semi-log scale ? And the colors of the X and Y axis are inconsistent. The y-axis is blue and x-axis is black. Why is this ?

If this is the mean profile from 6 rats at 5 mg/kg dose, can the authors provide standard deviation of the observed data and plot pharmacokinetics mean + sd on semi-log scale ?

Table 9: Can the authors be consistent with the casing of the table contents and decimal points ?

Line 302: change to further studied,

Round 2

Reviewer 1 Report

The revised mansuscript tries to reflect some reviewer's concerns and suggestions. However,major points is not cleared yet in the revised manuscript.

Regarding the previous formulation or delivery system studies for imperatorin, there are several literatures searched from Pubmed as below;

- Lin et al (2018), Increased skin permeation efficiency of imperatorin via charged ultradeformable lipid vesicles for transdermal delivery. Int J Nanomedicine.

- Pan et al (2010), Imperatorin sustained-release tablets: In vitro and pharmacokinetic studies, Arch Pharm Res.

The reviewer think that it is quite important that the pharmacokientics of imperatorin in proper vehicle should be generated with data and compared to authors' formulation in the mansucript.

Author Response

Response: Thank you for your suggestion; a supplementary pharmacokientics study of oral administration of imperatorin at a dose of 50mg/kg was carried out, and the results were compared to that of imperatorin Lipid Microsphere. We have added related content in the revised manuscript.

Reviewer 2 Report

Dear the Editor,

I guess the readability of revised manuscript pharmaceutics-379786 seemed improved from the first draft by extensive text-editing, thus it could be reasonable to consider this for publication.

Author Response

Thank you very much for your suggestion. In the introduction, we have provided sufficient background and include the relevant references.

Round 3

Reviewer 1 Report

The revised manuscript tried to reflect the reviewer's critical points/suggestions.